# Challenges to Cocoa Production in the Face of Climate Change and the Spread of Pests and Diseases

**Christian Cilas** [1,*] and **Philippe Bastide** [2,3]

1   CIRAD, DGDRS, University Montpellier, University FHB Abidjan, 01 BP 6483 Abidjan 01, Ivory Coast
2   CC&C, 34000 Montpellier, France; bastide.philippe@gmail.com
3   CIRAD, UMR ABSys, University Montpellier, TA B-34/02-Avenue Agropolis-34398,
    34398 Montpellier CEDEX 5, France
*   Correspondence: christian.cilas@cirad.fr

**Abstract:** The evolution of cocoa farming was quickly confronted with the development of pests and diseases. These sanitary constraints have shaped the geographical distribution of production over the centuries. Current climate change adds an additional constraint to the plant health constraints, making the future of cocoa farming more uncertain. Climate change is not only affecting the areas where cocoa is grown for physiological reasons, particularly in relation to changes in water regimes, but also affects the distribution of pests and diseases affecting this crop. These different points are discussed in the light of the trajectories observed in the different cocoa-growing areas. The breeding programs of cocoa trees for sustainable resistance to plant health constraints and climate change are therefore particularly important challenges for cocoa farming, with the other management practices of plantations.

**Keywords:** pest and disease management; climate change; cropping system; breeding for resistance

---

## 1. Historical Trajectory of Cocoa Production

When the Spaniards discovered Central America and Mexico (1504–1525), cocoa had already been produced, traded and consumed there for several centuries [1]. The cocoa tree was cultivated by the Aztecs and Mayas at that time, and archaeological studies have indicated its cultivation in the region called Soconusco (southern Mexico, northern Guatemala) long before that time [2]. Indeed, the first domestication of the cocoa tree would have been made by the Olmecs from 1000 BC. The Olmec civilization is considered the "mother civilization" of Mesoamerica [3]. Even more recent research indicates that the use of cocoa beans already existed in the upper Amazon (currently Ecuador and Peru) in 5000 BC [4]. However, the first organized plantations seem to have been carried out in the Soconusco region, sometimes with drainage and irrigation systems when rainfall was insufficient. The choice to plant cocoa trees in marginal rainfall conditions may be due to the identification of these areas as less prone to fungal diseases, such as pod rot due to various species of phytophthora. Indeed, traces of plantations date back to the centuries before the arrival of the Spanish; for example, cocoa tree fragments and drainage structures have been found at the archaeological site of Izapa in the state of Chiapas in Mexico [5]. Intensive plantations, with more than 1000 trees per hectare, are mentioned at the time of the conquistadors and the cocoa tree was often associated with *Gliricidia sepium* [6].

After the discovery of this plant and the use of cocoa by Europeans, the extension of its cultivation has continued to grow. A first extension of plantations took place throughout Central America in the 16th century. Then plantations developed in South America, particularly in Colombia, Ecuador, and Venezuela towards the end of the 16th century and the 17th century; a few plantations were also established in Southeast Asia at this time. From the 19th century onwards, the cocoa tree was massively

exported to other continents, mainly Asia and Africa (Figure 1). Cocoa is now cultivated in most humid tropical countries. The production cycles of the different producing countries have been described in numerous works [7,8], with stages of population migration, accompanied by deforestation, plantation establishment, then more or less rapid declines caused by socio-economic reasons, but also, as we will see, by the emergence of sanitary problems.

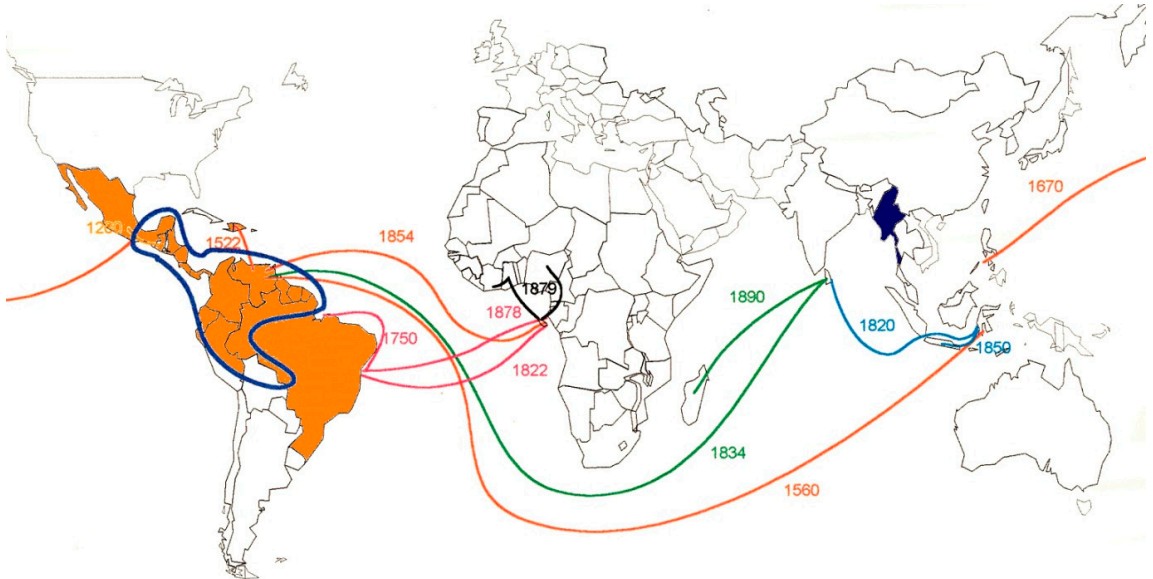

**Figure 1.** Expansion of cocoa cultivation from its area of origin, the American continent.

## 2. Sanitary Constraints and Geographic Distribution of Main Cocoa Pests and Diseases

Difficulties in cocoa tree cultivation were reported by the first Spaniards including, in particular, black pod rot (due to *Phytophthora* genus) when there is too much shade [9]. The health situation of cocoa trees has always been a major problem facing producers. Some pathogenic organisms, originating from the American continent, have co-evolved with the cocoa tree, while others originating from other producing continents have adapted to the cocoa tree after a host jump has occurred. A host jump means that an existing disease in a given species affects a new species that had not previously faced the disease in question.

With the expansion of cocoa cultivation to other continents, diseases that co-evolved with cocoa, such as frosty pod rot (*Moniliophthora roreri*) or witches' broom (*Moniliophthora perniciosa*), have so far spared the new producing continents. On the other hand, new diseases to which the cocoa tree had not previously been subjected have appeared (host jump), including cacao swollen shoot virus, *Phytophthora megakaria* in Africa, and vascular streak dieback in Asia. These diseases that have adapted to cocoa reveal the particular susceptibility of this plant species. Comparatively, the rubber tree (*Hevea brasiliensis*), which comes from the same area of origin, is less attacked by new diseases, although caution should be exercised in making predictions, particularly due to the influence of climate change, which influences epidemic risks [10].

Many pests, particularly insects, have also disrupted cocoa production in many geographical areas. Some insects exist in the areas where the cocoa tree originates, such as *Carmenta theobromae*, which is a pod borer [11]. Other insects have adapted to the cocoa tree, such as the cocoa tree mirids in Africa (*Salhbergella singularis* and *Distentiella theobroma*) or cocoa pod borer (*Conopomorpha cramerella*) in Southeast Asia. Diseases and pests have often been brakes that have limited the expansion of cocoa production in many parts of the world. The cocoa pod borer (CPB) has particularly hampered the development of cocoa production in most Asian countries. Malaysia, which produced up to 247,000 t in 1990, produces barely 1000 t today, mainly because of this pest [12,13]. Of course, CPB may not be

the only reason for this decline, but it was nonetheless a decisive constraint. Another example is how moniliasis drastically reduced cocoa production in Costa Rica from around 10,000 t in 1978 to around 600 t now [14]. Moniliasis is now in all the countries of Central America. In Mexico, cocoa production was around 50,000 t in 2003, just before the advent of moniliasis, and has decreased to under 30,000 t since 2007 [15]. For the moment, moniliasis has not moved to Brazil and Guyana, but the disease is in all the other producing countries of Latin America. In the Caribbean, just Jamaica is affected by moniliasis. Another example is production in the state of Bahia (Brazil), which decreased from 242,000 t in 1990 to 69,000 t in 1995, mainly due to the arrival of witches' broom in the state. The arrival of witches' broom would have been intentional in this case [16].

In Africa, Ghana became the world's largest cocoa producer in 1911, but ceded its position as the largest producer to Côte d'Ivoire in 1978, mainly due to the cocoa swollen shoot virus (CSSV) epidemic in Ghana [17]. The evolution of cocoa production areas is therefore clearly linked to the evolution of health pressures on the different production areas.

Until now, diseases originating in the areas of cocoa origin (moniliasis, witches' broom) have not spread to the other producing continents. Similarly, diseases that have adapted to cocoa in other continents, such as CSSV in Africa or vascular streak dieback (VSD) in Asia, have not left their continents of origin. However, extreme caution is needed to avoid the spread of diseases and pests between different production areas. The use of the quarantine facility at the University of Reading plays a key role in preventing the spread of diseases [18]. It is indeed strongly recommended that all transfers of plant material should go through this quarantine, even transfers within the same continent.

It is indeed important to prevent the spread of pests and diseases. For other crops, such as coffee, the lack of international quarantine has surely been a handicap to the prevention of the spread of pests and diseases. The coffee berry borer (*Hyptothenemus hampei*), which originated in Africa, is now present in almost all production areas [19].

## 3. The Effects of Climate Change

Global warming has been detected in most parts of the world and is expected to worsen in the future; its impact on agriculture has been studied for many years and models are being proposed for different crops [20]. Climate change can be characterized through different components; considering trees and climate change interactions, the main components influencing tree behavior are: (i) increasing atmospheric $CO_2$, (ii) warming, and (iii) changes in rainfall and drought patterns and duration [21]. Climate change is also changing climate events around the world. For example, the evolution of the El Niño–Southern Oscillation (ENSO) phenomenon, which has been studied for many years [22], shows more difficult transitions between rainy and dry seasons with extreme climatic events and consequences for crops and soils [23]. It was showed that the ENSO would probably cause decreased cocoa yields in the coming decades in cocoa agroforest of Bahia, Brazil [24].

Climate change will modify cultivation area characteristics and conditions for many crops and *Theobroma cacao* requirements, such as daily water needs [25], a high level of relative humidity, and the absence of wind [26,27], may not be met in several currently cultivated regions, such as West Africa [28]. Climate change, especially through water availability, $CO_2$ elevation, and highest elevation temperature, will induce important changes in terms of tree functioning and consequently phenology [29] and physiological traits such as water use efficiency, gaseous exchange, carbohydrate metabolism, and the translocation of assimilates and nutrients [30]. Several recent studies reviewed by Lahive et al. (2019) [27] show evidence of changes and interactions in cocoa tree behavior, linking genotypes, physiological traits, and climatic parameters.

According to Schroth et al. (2016) [31], climate change could induce a significant decrease in areas suitable for cocoa cultivation in West Africa (65% of current world production) due to drying, rather than increasing temperature (by about 2 °C by 2050), to which the cocoa tree seems less sensitive. Climatic drying would thus result in an increase in evapotranspiration resulting from an increase in temperature not compensated by annual rainfall, which is slightly decreasing, thus increasing water

deficit. This drying up would lead to reduced suitable land for cocoa, particularly due to the difficulties in setting up new fields, with significant mortality of young trees [32]. In such conditions, a lot of recent studies have aimed to promote smart agriculture to limit climate change impacts and have promoted agroforestry systems (AFSs) as a way to buffer those impacts and to reduce deforestation [33,34], even if an AFS cannot be an absolute response [35]. Climate change also has impacts on pollination and pollinators [36], and it would be useful to understand all the production components affected, from flowering to mature pods [37].

## 4. Climate Change, Global Change, and Plant Health Risks

Climate change may also alter plant health risks, depending on the diseases and pests present in different ecological zones [38,39]. The arrival of new pests and diseases is often due to human movement; the movement of goods and people is often blamed for the emergence of new health problems [40,41]. However, in Central America, the increase in the number of hurricanes, more frequent with climate change, and the associated strong winds were suspected in the rapid spread of moniliasis from Costa Rica to Mexico in the 2000s. The expansion of CSSV in Côte d'Ivoire seems to be linked to changes in the population dynamics of the vectors, which include several species of mealybugs. This change in population dynamics seems to be linked to climate change, or more generally global change, including ecological modifications linked to deforestation and its related impact on the protective buffer effect of tree strata [42,43]. In the future, some emergences and/or developments of pests and diseases could be due to climate change. Indeed, the distribution pattern of pests and diseases could change with changes in rainfall and temperature patterns and this aspect would indeed merit further study.

In view of these elements and based on the current, sometimes fragmentary, knowledge on the effects of climate change, rapid effects can be expected both in terms of the development and the intensity of health problems in cocoa plantations. Several breeding programs therefore have, as the main objective, the improvement of disease resistance. The aim is to build sustainable resistances in order to control certain plant health constraints as sustainably as possible.

## 5. Management in the Context of the Geographical Spread of Pests and Diseases and Climate Change

Several disease and pest control systems have been developed in line with the knowledge acquired, particularly in the fields of biology and chemistry. Biological control was certainly the first method used in agriculture to control plant health problems [44]. During the 20th century, advances in chemistry led to the development of pesticides to control various pests and diseases. The use of these pesticides then led to resistances, making these products less and less effective and requiring higher and higher doses. Moreover, the massive use of pesticides has led to human health problems, polluted environments, and most of the current research to control pests and diseases is aimed at getting rid of pesticides. Today, biological control is a subject of research to control several cocoa pests and diseases [45–47]. The development of more resilient cropping systems, including shade management, is also a promising research avenue to reduce biotic pressures in cocoa production [48–50]. Finally, the genetic improvement of cocoa trees for their resistance to pests and diseases is an essential lever to better manage cocoa sanitary problems [51–53]. However, pest and disease resistance are not the only selection criteria to be taken into account in genetic improvement processes. Productivity, quality, and adaptation to growing conditions, particularly important in the context of climate change, must also be improved. Multi-criteria selection is therefore necessary to develop varieties that meet the various objectives of the production chain. Selection indexes can be constructed for this purpose in order to jointly improve several characteristics [54,55], and molecular markers could be also developed in order to reduce breeding/selection cycles.

Adaptation to climate change is a more complicated concept because it requires determining the changes in climate that actually affect crops. Drying out or temperature elevation are a priori

the two main factors for which an evolution in agricultural management is possible. On the other hand, extreme climatic events (e.g., hurricanes or typhoons) are more complicated to take into account. Concerning the drying out of cropping areas or temperature increases, an adaptation of cropping systems is a possible response with the adaptation of adequate agroforestry systems [35]. The subject of research remains very open, as the choice of species to be associated with cocoa trees, as well as the distribution of species in production fields, still requires a great deal of research. In addition, certain sanitary constraints must be taken into account in the choice of tree species and plantation designs; for example, in Côte d'Ivoire and Ghana, where CSSV is prevalent, it is important to choose tree species that are not host to the virus, and it is important to surround plots with tree species that limit the spread of the virus [42,56]. Regarding the climate drying out, irrigation devices are sometimes used [25], and it should be remembered that irrigation was used for cocoa by the Aztecs and Mayas in pre-Columbian times [1]. Temperature elevation does not seem to pose significant problems for cocoa; however, a shift of cultivation areas to higher altitudes could be envisaged, particularly in Latin American countries that benefit from a wide variation in altitude. Genetic improvement is also an option, especially to adapt plant material to areas that are becoming drier, such as parts of Côte d'Ivoire or Ghana, the two largest cocoa producers.

## 6. Conclusions

Native from Latin America, the cocoa tree has been established in most intertropical areas, sometimes successfully, as in Côte d'Ivoire or Ghana, sometimes with failures due to pests that were particularly difficult to control, such as cocoa pod borer in Malaysia. The evolution of cocoa production has adapted to biotic and abiotic constraints that have changed over time under the influence of climate change and the emergence and development of pests and diseases (Figure 2). This history of cocoa farming should encourage us to anticipate risks, for example, through preventive breeding or the development of more resilient cropping systems. Research to establish new cocoa plantations in cultivation areas should also allow us to move away from the old model of "pioneer fronts", in which new plantations were established in cleared forest areas and their medium-term soil fertility was used. It is indeed important to preserve areas of tropical forest that have escaped deforestation and to propose new innovative agroforestry systems with adapted plant material for all component parts.

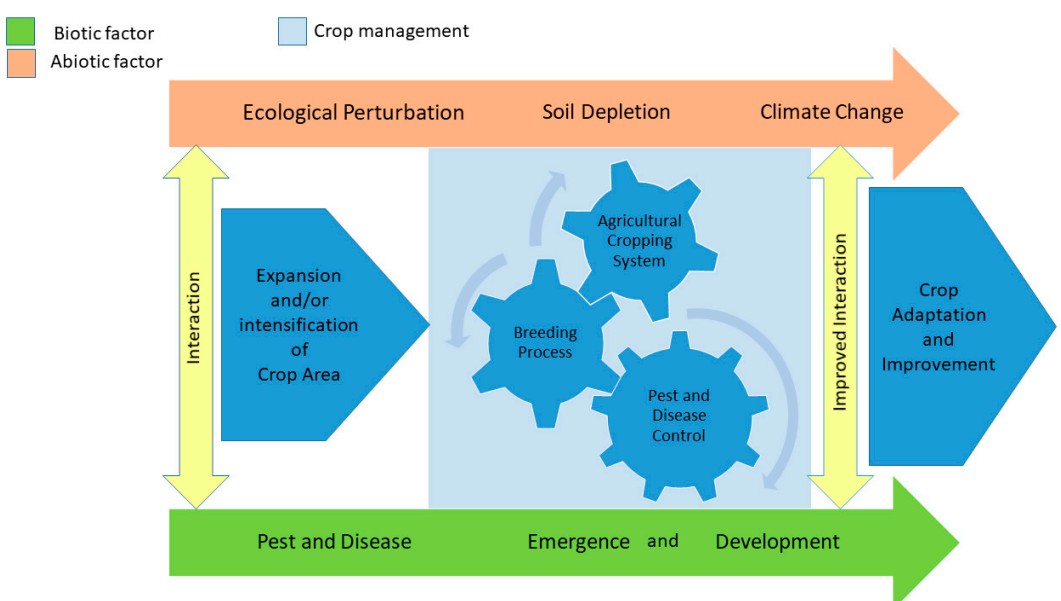

**Figure 2.** Evolution of cocoa cultivation, facing changing constraints.

**Author Contributions:** Conceptualization, C.C.; methodology, C.C. and P.B.; validation, C.C. and P.B.; formal analysis, C.C. and P.B.; investigation, C.C. and P.B.; resources C.C. and P.B.; data curation, C.C. and P.B.; writing—original draft preparation, C.C. and P.B.; writing—review and editing, C.C. and P.B.; visualization, C.C. and P.B.; supervision, C.C.; project administration, C.C. and P.B..; funding acquisition, C.C. All authors have read and agreed to the published version of the manuscript.

**Funding:** This research was funded by Cirad grant (DGDRS).

**Acknowledgments:** The authors hereby acknowledge Alain Rival for the invitation to write the paper for the Special Issue "Sustainable Production and Breeding Research for Tropical Plantation Crops".

**Conflicts of Interest:** The authors declare no conflict of interest.

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
