# Peer review of "Challenges to Cocoa Production in the Face of Climate Change and the Spread of Pests and Diseases"

_agronomy, doi:10.3390/agronomy10091232_

Round 1

Reviewer 1 Report

The authors revised most of the previous comments to improve the manuscript. I am still missing some points that should be addressed:

  1. The effect of the spread of pest and diseases by climate change is still poorly covered. This may include the distribution pattern of P+D due to shifts of rainfall patterns and temperature increase in regions where the pests where the pests were not present before.
  2. Line 11: write “geographical”
  3. Line 77-78: the sentence should be shifted between the two sentences in line 74 to start the new topic of yield reduction due to pests and diseases. The following sentence (line 78-79) can then be deleted.
  4. Line 76 – 90: As I mentioned before: use t or tons consequently.  
  5. Line 81 – 83: Consider “moniliasis reduced drastically cocoa production in CR from around 10,000 t in 1978 to around 600 t now”.
  6. Line 83-84: consider “Moniliasis is now in all the countries of Central America. In Mexico, cocoa production was around 50,000 t in 2003 before the advent of moniliasis in 2000, and decreased to under 30,000 t in …” (please add the correct year of the production amount)
  7. Line 122: Consider Läderach etal (2013) as a Reference for climate change effects on cocoa production in West Africa (Läderach et al. 2013: “Predicting the future climatic suitability for cocoa farming of the world’s leading producer countries, Ghana and Côte d’Ivoire“)
  8. Line 125: Relative humidity and Vapor pressure deficit are no physiological traits and should be deleted from the list.

Author Response

Responses :

  1. The effect of the spread of pest and diseases by climate change is still poorly covered. This may include the distribution pattern of P+D due to shifts of rainfall patterns and temperature increase in regions where the pests where the pests were not present before.  We added some elements. This aspect is also poorly documented in the literature. Indeed, it is often difficult to attribute the modification of the distribution of pests and diseases solely to climate change. Deforestation, large movements of people and goods are also important elements and it is often difficult to separate the effects of climate change from these other factors.
  2. Line 11: write “geographical” , done
  3. Line 77-78: the sentence should be shifted between the two sentences in line 74 to start the new topic of yield reduction due to pests and diseases. The following sentence (line 78-79) can then be deleted. done
  4. Line 76 – 90: As I mentioned before: use t or tons consequently.   done
  5. Line 81 – 83: Consider “moniliasis reduced drastically cocoa production in CR from around 10,000 t in 1978 to around 600 t now”. done
  6. Line 83-84: consider “Moniliasis is now in all the countries of Central America. In Mexico, cocoa production was around 50,000 t in 2003 before the advent of moniliasis in 2000, and decreased to under 30,000 t in …” (please add the correct year of the production amount) done
  7. Line 122: Consider Läderach etal (2013) as a Reference for climate change effects on cocoa production in West Africa (Läderach et al. 2013: “Predicting the future climatic suitability for cocoa farming of the world’s leading producer countries, Ghana and Côte d’Ivoire“) done
  8. Line 125: Relative humidity and Vapor pressure deficit are no physiological traits and should be deleted from the list. done

Reviewer 2 Report

The authors have addressed all major concerns.

Author Response

Thanks  for the comments

This manuscript is a resubmission of an earlier submission. The following is a list of the peer review reports and author responses from that submission.

Round 1

Reviewer 1 Report

  1. The topic is interesting and the manuscript well written. The authors address an important topic (pest and diseases) in cocoa climate change adaptations. The historical perspective of how pest and disease have contributed to shaping the current geographical distribution of global cocoa production is very insightful.
  2. Suggestion for further further improvement: In section "4. The effect of climate change", the discussion are quite general and would be helpful if it can be focused. Interesting question will be how the key environmental changes related to climate change  (LN 113, i), ii) and iii) can influence the dynamics of the major pest and diseases in the various geographic regions. e.g potential impact of cocoa climate suitability changes in West Africa as presented by Schroth et al 2016, 17 on CSSVD.
  3. Additional interesting reference for section 6 (https://www.sciencedirect.com/science/article/abs/pii/S0167880917304310?via%3Dihub)

Reviewer 2 Report

General comment

The article is reviewing the important topics climate change and pests and diseases that are challenging cocoa production worldwide now and with a future perspective. Both topics have a huge influence on cocoa production independently and in combination. The title promises a review on the effects of Climate change on cocoa production and of pests and diseases on cocoa production and of the combined effect, and the manuscript unfortunately is not providing this satifsfactorly. Also the combination of both, e.g. how climate change may affect the spread of pests and diseases by improving their environmental conditions, that should be the heart piece of this study, is not covered in depth and has to be improved.

Specific comments:

Line 11: I guess you refer to “geographical distribution” instead of only “geography”

Line 17-18: revise the sentence. Delete “programmes” if you refer to breeding in general. Or change “is” to “are”.

Line 40: Ponce 1586 in missing in the reference list

Line 41: “use” instead of “uses”

Line 43-46: avoid repeating “then .. then…”

Figure 1: may you add distribution areas of main cocoa pests and diseases? Would be more interesting for the topic of the article and a conclusion of the chapter 2 showing the coevolution areas and the host-jump areas.

Line 54: of main cocoa pests and diseases

Line 55-56: make clear, what disease you are talking about (black pod, to avoid confusion with frosty pod rot), species or at least name of the genus.

Line 57-78: Avoid repeating information; better combine the first paragraph with the following two.

Line 63: species names of monilisasis (frosty pod rot) and witches’ broom

Line 65, 67, 74, 121…: species names in italics.

Line 77-82: the information you are giving about Malaysian production decrease are doubled, but the numbers are not the same. Which are the correct production data? Write t or tons, but constantly.

Line 88: reference for Mexican production missing. What about Moniliasis in the other Latin-American countries?

Line 94-95: avoid repetition

Chapter 3: the first part is only a repetition of the before mentioned diseases, the last paragraph about coffee, the whole chapter is redundant; the information about Quarantine can be combined with the previous chapter.

Chapter 4: add information on the main climate change effects on cocoa trees and cocoa production and in which areas they are occurring and will affect more.

Line 115: ENSO is not actually a climate change result, but effects may be stronger by CC. keep causalities in mind.

Line 115: what evidences? What are harder transitions?

Line 121: explain the cocoa requirements

Line 122: temperature increase? Also increase of temperature extremes (high and low temperatures without changing the mean temperature) maybe harmful for cocoa.

Line 122: precipitation is not mentioned here, but water availability will be as or more important than temperature, as you mentioned also, especially as you mention WUE and gas exchange as consequences.

Line 124: RH is not a physiological trait. You don’t have to give the abbreviations if you are not using them in the following.

Line 129: ”cocoa”

Line 130: delete “in”

Line 131: Climatic drying

Line 131-133: unclear, rewrite. Remind the causality from temperature increase – higher ET, more problematic due to water scarcity.

Line 134: write clear that the seedling mortality is a result of drought.

Line 134: replace “et” by “and”

Line 138-139: this belongs already to the next chapter. And explain it.

Chapter 5: this should be the most interesting chapter of the manuscript describing the effects of CC on pests and therefore on cocoa: trees more susceptible because stressed by CC; pests / diseases can spread by weather extremes; CC makes some areas more suitable for the pests?

Line 143-144: This is not part of CC, not part of this chapter

Line 149: not clear why deforestation is mentioned here, not related to CC; keep causalities clear, what is direct effect of CC and what is indirect; reference missing

Chapter 6: especially the beginning is very general, write more directly related to cocoa (biological control). Describe selection criteria for genetic improvement for cocoa cultivars to CC; describe how resilient cropping systems can help cocoa production for CC or P+D.

Figure 2: Not clear at all: Interaction of crop management factors, which factor is activating which and why? Crop adaptation is not a consequence from pest control. Biotic/Abiotic factors: do the arrows indicate a direction?

Line 216: delete >>

Line 232: (2016)

Line 254: delete b in “…orbBlack pod…”